

# Climate projections of a multi-variate heat stress index: the role of downscaling and bias correction

Ana Casanueva[1], Sven Kotlarski[1], Sixto Herrera[2], Andreas M. Fischer[1], Tord Kjellstrom[3,4] and Cornelia Schwierz[1].

[1]Federal Office of Meteorology and Climatology MeteoSwiss, Zurich, Switzerland
[2]Meteorology Group, University of Cantabria, Santander, Spain
[3]Centre for Technological Research and Innovation (CETRI), Limmasol, Cyprus
[4]Australian National University, Canberra, Australia

*Correspondence to*: Ana Casanueva (ana.casanueva@meteoswiss.ch)

**Abstract.**

Along with the higher demand of bias-corrected data for climate impact studies, the number of available data sets has largely increased in the recent years. For instance, the Inter-Sectoral Impact Model Intercomparison Project (ISIMIP) constitutes a

framework for consistently projecting the impacts of climate change across affected sectors and spatial scales. These data are very attractive for any impact application since they offer worldwide bias-corrected data based on Global Climate Models (GCMs). Complementary, the CORDEX initiative has incorporated experiments based on regionally-downscaled bias-corrected data by means of debiasing and quantile mapping (QM) methods. In light of this situation, it is challenging to distil the most accurate and useful information for climate services, but at the same time it creates a perfect framework for

intercomparison and sensitivity analyses.

In the present study, the trend-preserving ISIMIP method and empirical QM are applied to climate model simulations that were carried out at different spatial resolutions (CMIP5 GCM and EURO-CORDEX Regional Climate Models (RCMs), at approximately 150km, 50km and 12km horizontal resolution, respectively) in order to assess the role of downscaling and bias correction in a multi-variate framework. The analysis is carried out for the wet bulb globe temperature (WBGT), a heat

stress index that is commonly used in the context of working people and labour productivity. WBGT for shaded conditions depends on air temperature and dew point temperature, which in this work are individually bias-corrected prior to the index calculation. Our results show that the added value of RCMs with respect to the driving GCM is limited after bias correction. The two bias correction methods are able to adjust the central part of the WBGT distribution, but some added value of QM is found in WBGT percentiles and in the intervariable relationships. The evaluation in present climate of such multivariate

indices should be performed with caution since biases in the individual variables might compensate, thus leading to better performance for the wrong reason. Climate change projections of WBGT reveal a larger increase of summer mean heat stress for the GCM than for the RCMs, related to the well-known reduced summer warming of the EURO-CORDEX RCMs. These differences are lowered after QM, since this bias correction method modifies the change signals and brings the results



for GCM and RCMs closer to each other. We also highlight the need of large ensembles of simulations to assess the feasibility of the derived projections.

# 1 Introduction

In the last years the amount of available climate projection data has largely increased thanks to the development of
intercomparison projects (Coupled Model Intercomparison Projects CMIP, Taylor, et al., 2011; Inter-Sectoral Impact Model Intercomparison Project ISIMIP, Warszawski, et al., 2014) and other initiatives (CORDEX, Giorgi, et al., 2009; Jones, et al., 2011; CORDEX-Adjust). Due to this, there have been many efforts towards the distillation of climate data into usable climate information (Hewitson, et al., 2014; Fernández, et al., 2018). This is largely hampered by the large envelope of uncertainty, which grows in the subsequent steps in the production of climate information, the so-called "uncertainty
cascade" (Wilby & Dessai, 2010). In this work we assess the role of downscaling and bias correction as key elements of the development of climate information. For this purpose, we intercompare climate change projections of heat stress in Europe coming from different data sources, at different spatial resolution and corrected with two different bias correction methods in order to identify the major sources of uncertainty in terms of present and future climate.

Global Climate Models (GCMs) are able to reproduce the main features of the climate system and are commonly used to
examine changes in climate on a global scale (Taylor, et al., 2011). Despite the recent improvements, systematic biases remain and the model resolution is still too coarse to adequately describe mesoscale processes (Giorgi & Mearns, 1991). Regional Climate Models (RCMs) are frequently used to bridge the gap between the GCM and the regional-to-local scales (Giorgi, 2006; Feser, et al., 2011). They solve the governing equations of the climate system in a limited spatial domain using initial and boundary conditions from GCMs (reanalysis for the model-evaluation experiments). Despite the increased
horizontal resolution, RCMs, similar to GCMs, include physical parameterizations for subgrid processes which occur at spatial scales smaller than the model grid spacing (microphysics, convection, radiation, etc.). RCMs add valuable information with respect to their driving GCM due to more detailed spatial patterns and the better representation of local processes, e.g. high precipitation frequencies (see e.g. Maraun, et al., 2010; Warrach-Sagi, et al., 2013). However, both GCMs and RCMs are prone to systematic biases and some sort of bias adjustment or correction is typically needed before
they are used in impact modelling (Christensen, et al., 2008; Hagemann, et al., 2011). Bias correction (BC) methods typically adjust some features of the model distribution (e.g. the mean or percentiles) towards the observed counterparts, partly removing systematic errors in the model output. However, the added value of bias-corrected RCM simulations with respect to the bias-corrected GCM counterparts remains unclear. The same question applies for the difference between bias-corrected high resolution RCM simulations (at approximately 12km) and the coarser counterparts (at approximately 50km).
For the latter, Casanueva, et al., 2016 showed that the added value (in terms of mean, percentiles and precipitation frequency) is not statistically significant after applying simple (scaling) bias correction methods.





Many BC methods with different characteristics have been described in the literature (Maraun, et al., 2010; Piani, et al., 2010; Gutiérrez, et al., 2018): empirical or parametric methods, variable-specific (e.g. assuming a certain distribution) or non-specific methods, multi-variate or univariate methods, seamless or for specific time horizons (e.g. correction of ensemble spread in monthly/seasonal forecasts). All of them consist of a training phase (in which the correction function is

calibrated) and an application phase under different conditions. Note that the correction functions (calibrated in present climate) are assumed to be invariant on time (stationarity assumption). Moreover, in a climate change context, the way the correction is applied in future climate might affect the climate change signal.

A specific bias correction method was developed in the framework of the ISIMIP initiative (Warszawski, et al., 2014). This project attempted to offer a consistent framework for cross-sectoral, cross-scale modelling of the impacts of climate change

in order to ease the application of climate model data and meet user-specific needs. The ISIMIP method (Hempel, et al., 2013; ISIMIP2b, Frieler, et al., 2017) was applied to several GCMs from the CMIP5 (5th Phase of Coupled Model Intercomparison Project; Taylor, et al., 2011). The ready-to-use, bias-corrected data have been used to produce impact model simulations for different sectors such as agriculture, biomes, forests, fisheries permafrost, etc. as well as to derive climate impact indices, including heat stress (Kjellstrom, et al., 2018).

Among other BC methods, empirical quantile mapping stands out as one of the most widely used methods. Despite its limitations and shortcomings (Maraun, et al., 2017; Lanzante, et al., 2018), it is one of the best performing bias correction and statistical downscaling methods in evaluation experiments (Gutiérrez, et al., 2018; Hertig, et al., 2018). One reason for this might be that it is often favored by the evaluation metrics -commonly based on moments of the probability density function- considered in the intercomparison experiments. Quantile mapping is, by construction, able to correct for intensity-

dependent biases. As a consequence, it can modify the raw model climate change signal, which might be debatable (Gobiet, et al., 2015; Casanueva, et al., 2018a; Ivanov, et al., 2018). In contrast, the main objective of the ISIMIP correction is to preserve the trend of the raw data in the calibration period.

In the present work, we consider CMIP5 and EURO-CORDEX (European branch of CORDEX) simulations, the later at the two available spatial resolutions (approximately 12 km and 50 km), and the ISIMIP correction and empirical quantile

mapping as bias correction methods:

- To assess the added value of a more complex (in terms of the number of parameters calibrated) bias correction method (empirical quantile mapping) with respect to the ISIMIP correction,

- To assess the added value of RCMs compared to their driving GCM after bias correction, and

- To assess the impact of downscaling and bias correction in the climate change signal.

The added value is examined by evaluating several statistics under present climate conditions and exploring the feasibility of climate change projections. All analyses are applied in the context of climate change projections of heat stress in Europe. Heat stress depends mainly on temperature and humidity (low wind speed and high solar radiation also contribute to heat stress but are not considered in this work). Whereas the empirical quantile mapping is a univariate bias correction method (typically with the same core implementation for all variables when it is applied in a multi-variate context), the ISIMIP



correction includes dependencies between some variables (e.g. mean temperature is needed to correct maximum/minimum temperatures) in order to preserve the physical consistency among them. Hence, the ability of the methods to reproduce multi-variate structures is implicitly investigated.

## 2 Data and Methods

### 2.1 Heat stress index

Under very hot and humid conditions, the ability of the human body to regulate the core temperature and dissipate heat via sweat evaporation is reduced, provoking heat stress (Koppe, et al., 2004; Parsons, 2014). Other meteorological variables such as strong radiation or low wind speed can exacerbate heat stress. Such conditions directly affect human well-being and can develop into heat-related illnesses such as fatigue, muscle cramps and heat stroke. In the context of working people,
several studies revealed the negative impact of heat stress on workers' health (Pogačar, et al., 2018) and labour productivity (Kjellstrom, et al., 2009; Ioannou, et al., 2017). International organizations such as the International Standards Organization (ISO) and the US National Institute for Occupational Safety and Health (NIOSH) have developed guidelines to protect working people against heat stress (ISO, 1989; ISO, 2017; NIOSH, 2016). The recommendations comprise work-rest cycles and water intake under specific heat conditions. A combination of technical, regulatory and behavioural measures is needed
to adapt workers to increasing temperatures at an individual, sectoral and governmental level (Vivid Economics, 2017). In the context of global warming, the development and dissemination of heat-health planning and warning systems is now among the priorities of the World Meteorological Organization (WMO) and the World Health Organization (WHO; WMO, 2015), as well as the International Labour Organization (UNDP/ILO, 2016) and the International Organization for Migration (IOM, 2016). Within this framework, the Horizon 2020 HEAT-SHIELD project (www.heat-shield.eu) aims to address the
effects of climate change on European working population within an inter-sectoral framework (Nybo, et al., 2017). In particular, one of the specific objectives of the project is to generate climate change projections of heat stress (Casanueva, et al., 2018b).

There are many indices based on meteorological variables which have been often used to assess occupational heat stress conditions in the literature (de Freitas & Grigorieva, 2015; Coccolo, et al., 2016). The wet bulb globe temperature (WBGT)
has been chosen in the HEAT-SHIELD project as primary heat stress index since it can be computed from standard meteorological variables available in both the observations and climate models and it can be interpreted by occupational scientists and physicians by means of the corresponding international (ISO) and national (e.g. NIOSH, 2016) standards regulations, and adjusted according to the workers' clothing.

In this study we focus on the WBGT in the shade (Bernard & Pourmoghani, 1999; Lemke & Kjellstrom, 2012), which
assumes that there are no strong radiation sources (the globe temperature equals the air temperature) and wind speed of 1m/s, which corresponds to the movement of arms or legs during work. Bearing this in mind, the input variables for the calculation of the WBGT in the shade are air temperature and dew point temperature. The latter accounts for the humidity conditions



and can be obtained from daily mean temperature and relative humidity (or specific humidity and air pressure) in models and observational data sets. In order to account for the highest daily heat stress, we used daily maximum temperature and daily mean dew point temperature (unlike relative humidity, usually it only slightly varies along the day) to approximate the daily maximum WBGT. WBGT is calculated through an R package *HeatStress* (https://github.com/anacv/HeatStress).

## 2.2 Observational data

The observational reference used to validate the climate models and perform the bias correction is the WFDEI (WATCH Forcing Data methodology applied to ERA-Interim; Weedon, et al., 2014) data set, which is based on the ERA-Interim reanalysis (Dee, et al., 2011) corrected by the CRU observational data set (or GPCC for precipitation). It is developed on a 50km regular grid and provides 3-hourly and daily values of temperature, precipitation, humidity, wind and radiation, among others. Its predecessor, WFD was used as observational reference in the ISIMIP 2a experiment (Hempel, et al., 2013) and its successor (EWEMBI, Frieler, et al., 2017) in the newer ISIMIP2b. Note that the WFDEI data are identical to the EWEMBI over land and for the considered variables (daily maximum and mean temperature, specific humidity and surface air pressure). In the present work we use the WFDEI data set over Europe for the period 1981-2010. Daily maximum temperature is obtained as the maximum of the 3-hourly values.

## 2.3 Global and Regional Climate Model data

The GCM considered in the present analysis is the HadGEM2-ES_r1i1p1 (Collins, et al., 2011; denoted as HadGEM along the paper) from CMIP5, which is one of the GCMs used in the ISIMIP experiments. Data covering the EURO-CORDEX domain were extracted considering only land grid boxes (land area fraction larger than 50%).

We additionally use the EURO-CORDEX RCM simulations (Jacob, et al., 2014; Kotlarski, et al., 2014) driven by HadGEM2-ES accessible via the Earth System Grid Federation (ESGF archive, https://esgf.llnl.gov) as of May 2017. These are the same HadGEM-driven RCM simulations as used by Casanueva, et al., 2018b, in a comprehensive study about climate projections of heat stress in Europe. The RCM simulations were conducted at two different spatial resolutions which correspond to approximately 12 (EUR-11) and 50 km (EUR-44) grid spacing. The final set of regional models consists of RACMO, CCLM and RCA run by the KNMI (Royal Netherlands Meteorological Institute), CLMcom (Climate Limited-area Modelling Community) and SMHI (Swedish Meteorological and Hydrological Institute), respectively. The historical simulations cover a common historical period 1981-2005 (the five years 2006-2010 from the scenario simulations are added in this work to complete the observational period) and future projections cover the period up to 2099. The available RCPs (Representative Concentration Pathways) vary for each GCM-RCM combination (Table 1).

We retrieved GCM and GCM-RCM data for daily maximum temperature, as well as daily mean temperature and relative humidity (or specific humidity and sea level or surface air pressure, depending on the model) that were used to calculate daily mean dew point temperature. Note that HadGEM as well as the HadGEM-driven RCMs present a 360-day calendar so, to harmonize this with the observations, 5 (or 6 in leap year) missing values were included randomly along each year, but



keeping the same position for all variables (to avoid inter-variable modifications), RCPs and models. All analyses were carried out at the spatial resolution of the observational grid (regular 50x50km). For this reason, all model simulations (GCM and GCM-RCMs) were conservatively remapped into the WFDEI grid (1st order conservative remapping, as in the ISIMIP experiments). As a consequence, there will be aspects of the added value of the high-resolution EUR-11 experiments (related

to better-resolved, fine-scale processes; Prein, et al., 2015) that can be smoothed out, but some may still be present after remapping them onto a coarse resolution.

[Table 1]

**Table 1: EURO-CORDEX RCMs driven by HadGEM, for the two spatial resolutions (EUR-11 for approximately 12km spatial**
**resolution and EUR-44 for approximately 50km resolution) and three RCPs (RCP2.6, RCP4.5 and RCP8.5).**

## 2.4 Bias correction methods

### 2.4.1 ISIMIP bias correction

The ISIMIP bias correction was developed in the framework of the ISIMIP project (Hempel, et al., 2013). It consists of a
correction of the monthly mean biases followed by the correction of the daily variability around the monthly mean. For temperature the monthly correction is additive, whereas it is multiplicative for precipitation, radiation and wind. The daily variability correction consists of a parametric quantile mapping adjusting a normal distribution for temperature and a gamma distribution for precipitation. After ISIMIP (see light dashed green line in Fig.1a), the mean of the historical data is adjusted towards the observations (black lines) but the variance and shape of the raw distribution is mostly retained. The monthly
means and monthly variability are adjusted using only a constant correction (either an offset or a multiplicative factor) in the historical and future periods (see green lines in Fig.1a for an example for temperature; light and dark green lines represent the historical and future bias-corrected data through ISIMIP, respectively). Therefore the corrections cancel out when calculating the mean (additive or relative) climate change signal and the long-term trend of the raw simulated variables (red arrow in Fig.1a) is preserved.
The ISIMIP correction includes dependencies between some variables (e.g. mean temperature and wind speed are needed to correct maximum/minimum temperatures and eastward/northward wind components, respectively) in order to preserve the physical consistency among them. However, there are not implemented dependencies between temperature and relative humidity yet. This BC method correction is implemented for several variables as part of the R package *downscaleR* (Bedia, et al., 2017), included in the R bundle *climate4R* (Cofiño, et al., 2018; Iturbide, et al., 2019). We correct dew point
temperature following the same procedure as for daily mean temperature, thus, dependencies with other variables are not considered. As mentioned before, daily mean temperature is used in the correction of the daily maximum temperature in



order to maintain the physical consistency between variables. Although the ISIMIP initiative provides bias-corrected GCM data, for the sake of consistency we apply the corrections to the raw GCM, as well as RCM data.

### 2.4.2 Empirical quantile mapping (QM)

In this work we use the implementation from Déqué, 2007; Rajczak, et al., 2016 which consists of the correction of the 99 percentiles of the empirical distribution of the model towards their observational counterparts. The corrections between two consecutive percentiles are linearly interpolated and constant extrapolation is considered for the values beyond the calibration range, i.e. the correction of the 99th (1st) percentile is applied to values above (below) the calibration range, (Themeßl, et al., 2012). The correction is calibrated for each day of the year with a 91-day moving window. It is a univariate BC method and in this work it was applied independently to daily maximum temperature and daily mean dew point temperature.

For a historical simulation (see e.g. light dashed purple lines in Fig.1a) the corrected data largely resemble the distribution of the observations. During the application of QM to a future climate simulation, the model data are mapped into the percentiles of the training data and the corresponding correction function is applied (dark purple lines in Fig.1a), thus QM would correct differently the future and the historical distributions if the relative frequencies in the future differ from the training counterparts (Casanueva, et al., 2018a). Therefore, QM is able to correct for intensity-dependent biases and, subsequently, modifications of the raw model climate change signal may occur. In the example for temperature in Fig.1a, QM narrows the distribution of the future simulated data, thus leading to a smaller mean change signal than the raw counterpart (see purple and red arrows).

### 2.4.3 Application of the bias-correction methods

For both bias correction methods, the corrections are applied independently to each grid box of each GCM/RCM, resolution (if applicable) and RCP. These corrections are calibrated in the period 1981-2010 and are applied (1) to the same period to evaluate the performance in present climate and (2) to a future period at the end of the 21st century (2070-2099) to produce bias-corrected climate projections. Due to the multi-variate nature of the WBGT, we correct separately daily maximum temperature and daily mean dew point temperature prior to the WBGT calculation (i.e. component-wise approach, Casanueva, et al., 2018a). Although the BC methods are applied to the full time series (monthly mean correction for ISIMIP, 91-day moving window centred on each day of the year for QM), all results shown refer to the summer season (June, July, August), since it is the time when extreme heat stress conditions occur.

As shown in Figure 1b, the ISIMIP bias correction is applied to the GCM and the RCM to assess the added value of the RCMs after bias correction. We additionally correct the climate models using quantile mapping to assess the added value of a more complex (in terms of the number of parameters calibrated) bias correction method.

[Fig.1]





**Figure 1: (a) Example of the effect of the two bias correction methods on the empirical cumulative distribution function (left) and the probability density function (right). Observations are depicted in black and historical (HIST) and future (FUT) model simulations in light and dark colours, respectively. Raw data are depicted in red, ISIMIP-corrected data in green (upper panel) and QM-corrected data in purple (lower panel). The magnitude of the mean change signal is shown with the arrows. This example corresponds to daily maximum temperature as represented by HadGEM for an exemplary grid box (HIST: 1981-2010, FUT: 2070-2099 for RCP8.5). (b) Conceptual scheme of the present study.**

## 3 Results

### 3.1 Evaluation of mean biases of WGBT

The two BC methods (ISIMIP and QM) are applied to the two primary variables of the heat stress index, namely daily maximum air temperature and daily mean dew point temperature, prior to the WBGT calculation. Under no cross-validation (i.e. the methods are calibrated and validated in the same period) both BC methods adjust, by construction, the central part of the distribution (mean for ISIMIP, median for QM, see Fig.1a). ISIMIP further adjusts the variability around the mean, whereas QM additionally adjusts the 99 empirical percentiles. The performance in terms of mean biases of the two BC methods for individual variables is good and differences related to the parametric (ISIMIP) or empirical (QM) nature of the method may arise on the tails of the distribution, where QM outperforms ISIMIP (not shown).

The suitability of the component-wise BC approach of the WBGT prior to its application in a climate change context is assessed by evaluating the corrected WBGT with the observed counterpart for the period 1981-2010. Although the calibration and validation periods are the same, our approach can be considered independent since the evaluated aspect (i.e. multivariate consistency and WBGT statistics) is not directly tackled by the BC methods. An additional split-sample cross-validation (cold vs. warm years, not shown) indicates that WBGT biases are of the same order of magnitude as in the non-cross-validated analysis.

Mean biases of summer mean WBGT (Fig.2, upper panel) are evident for the raw GCM and RCMs (blue boxes; note that no height correction has been applied to the raw data, which might be the main responsible for the skewed distribution of the raw biases). These are largely reduced after both BC methods, equally well for GCM and RCM data. The evaluation of the 95th and 99th percentiles reveals better performance for ISIMIP or QM depending on the model. QM improves on mean biases and reduces their variability in higher percentiles for the GCM, RACMO and RCA, whereas a cold bias emerges for CCLM. There is no evident added value of the RCMs with respect to the GCM after bias correction (see also the spatial pattern of the differences between bias-corrected GCM and RCMs in Fig.S1). This is in agreement with the findings for coarser vs. higher resolution of RCM simulations by Casanueva, et al., 2016.

[Fig.2]

**Figure 2: Biases of mean (first row), 95[th] percentile (second row) and 99[th] percentile (third row) of summer WBGT for the GCM and RCMs at EUR-44 and EUR-11. Biases are calculated as model minus observations. Each box represent the biases across all grid boxes for the raw (blue), ISIMIP-corrected (orange) and QM-corrected (green). Due to the different land-sea masks in the observations, GCM and RCMs (EUR-44 and EUR-11), all boxplots consider the grid boxes common to all data sets.**



The spatial pattern of biases in the 99th percentile of the WBGT (WBGTp99) is shown in Fig.3. In general, the bias correction methods alleviate the biases of the raw models over Europe (in particular, large biases due to complex orography), although there are cases where, in some regions, the biases after bias-correction remain as high or even higher than for the raw output. For the GCM, biases of similar magnitude remain after ISIMIP and QM, with completely different spatial structures. For the RCMs RACMO and RCA, slightly better results are found for QM compared to ISIMIP, with biases up to ±1°C. The added value of the two BC methods with respect to the raw simulations is also shown in Fig.S2, being larger in areas with complex orography and slightly better for QM. The above mentioned cold bias for the CCLM after QM is present especially in eastern Europe (Fig.3m and S2h). The causes for that are analysed in more detail in the next section.

[Fig.3]

**Figure 3: Spatial distribution of the observed 99th percentile of summer WBGT (WBGTp99, panel a) and model biases for the GCM (b-d) and RCMs-EUR11 (e-m), for the raw (first column), ISIMIP-corrected (second column) and QM-corrected (third column). The grid boxes over Warsaw and Madrid are marked as reference for subsequent analyses.**

## 3.2 Evaluation of intervariable relationships

The component-wise correction of the WBGT is able to correct for large biases in some WBGT statistics, but some biases remain for specific locations and models. We focus on the eastern European region and select the closest grid box to the city of Warsaw (Poland), where the original positive bias of WBGTp99 (0.7°C) turns into a negative bias (-1.4°C) after QM for CCLM-011 (Fig.3m). The application of QM to the input variables of WBGT reduces their raw biases to less than ±0.3°C for all analysed statistics in that grid box. QM corrects for distributional biases of each variable, but the temporal sequence (i.e. day-to-day variability) of the raw data is not altered and the ranks are preserved. Given that maximum temperature and dew point temperature are combined non-linearly to produce the daily sequence of WBGT, deficient intervariable relationships may lead to an inaccurate representation of pairs of input variables and, consequently, to biases in the WBGT distribution. We assess pairs of values of maximum temperature and dew point temperature that produce the highest values of WBGT (in particular those above the 95th percentile, WBGTp95) for the observations and model (raw and bias-corrected) data (Fig.4). According to the observations, the 5% of days with the highest heat stress is produced by high maximum temperatures (28-36°C) and high dew point temperature (13-21°C). Within these ranges, WBGT can reach values of 23-27°C (see circles in Fig.4). The raw models (squares in Fig.4a,d) present some biases on the upper tail of the distribution of the two input variables, which translates in a positive biases of the WBGTp99 in the two models. Raw RACMO (Fig.4a) overestimates maximum and dew point temperatures but captures rather well the intervariable relationships, whereas raw CCLM (Fig.4d) presents more deficiencies in representing the intervariable structure and, in particular, shows large positive biases for maximum temperature and negative biases for dew point temperature. Overall, the remaining biases after the




ISIMIP correction (downward triangles in Fig.4b,e) approximately resemble the original counterparts for RACMO and improve on the raw data for CCLM, whereas QM (upward triangles in Fig.4c,f) overcorrects the original biases. For CCLM-QM the highest 5% WBGT values are produced by lower values of both input variables compared to the observed pairs, especially dew point temperature (down to 5°C) leading to an underestimation of the WBGTp99. Such low dew point temperatures are also found for CCLM-ISIMIP, but they are combined with positively biased maximum temperatures, thus the biases (too high maximum temperatures -above 38°C- and too low dew point temperatures -below 12°C- , see top left corner in Fig.4e) compensate leading to a small bias in WBGTp99. Therefore the evaluation of WBGT statistics should be done with caution since results can be right for the wrong reason, highlighting the need of multi-variable model evaluations (García-Díez, et al., 2015).

[Fig.4]

**Figure 4: Intervariable relationship for the observations and (raw and bias corrected) model data, for RACMO-011 (a-c) and CCLM-011 (d-f) for the grid box over Warsaw. Each scatter plot represents pairs of values of daily dew point temperature (X-axis) and maximum temperature (Y-axis) which produce summer WBGT values above WBGTp95. The three coloured markers correspond to WBGT values for the observations (circles in all panels), raw RCMs (squares in a,d), RCM-ISIMIP (downward triangles in b,e) and RCM-QM (upward triangles in c,f). Isolines also represent WBGT values and the thicker line depicts the observed WBGTp99.**

To investigate in more detail the effects of the downscaling methods on the full joint probability distribution of the maximum temperature and dew point temperature, consider Fig. 5. It shows the 2-dimensional kernel density distribution together with the marginal histograms for the same grid box (Warsaw) as in Fig. 4. Higher values of the observed joint probability (Fig.5, top panel) are associated with more likely values of maximum temperature and dew point temperatures, around the mean of the distribution (approximately 23°C and 10°C, respectively). For the GCM, the distributions of the WBGT input variables are wider than the observed ones, leading to a more diffuse and displaced distribution of joint probabilities (Fig.5, second row). In agreement with the previous results, after ISIMIP the joint probabilities are centred, but neither the shape nor the maximum values are well represented. QM systematically narrows the distributions and slightly improves the results, which is consistent with the higher values of the Perkins score (Perkins, et al., 2007). The raw RACMO and RCA outputs tend to represent better the shape of the joint distribution than the GCM, although the maximum probabilities are biased towards somewhat lower maximum temperatures for RACMO as well as towards lower maximum temperature and higher dew point temperature for RCA. The ISIMIP correction largely preserves the original structure in the raw data, whereas QM often narrows the original, skewed distributions towards the observed counterpart. There is, however, an overestimation of the maximum probabilities after QM. The CCLM raw simulations (EUR-44 and EUR-11) for this grid box present more deficiencies in representing the intervariable structure, in terms of the magnitudes and location of the joint probabilities. The ISIMIP correction brings the CCLM maximum closer to the observational counterpart, but the joint probabilities are too wide. For instance, unlike the observations, there is some probability of high values of WBGT (see isoline denoting observed WBGTp99) associated to rather low dew point temperatures and high maximum temperatures (as



also shown in Fig.4). This problem is very likely inherited from the raw data, and is slightly improved by QM. The remaining underestimation of WBGTp99 after QM is also visible from this plot, since the probability above the observed WBGTp99 isoline is negligible. In terms of the general structure, the joint distributions of the RCM-QM data are better than those with the ISIMIP correction, although the performance greatly depends on the quality of the raw data.

RACMO (especially EUR-44, not shown) is the best performing model in terms of joint probabilities for this specific grid box, with slightly improved results after QM. The improvement of QM on the joint probabilities is more noticeable in RCA, CCLM and the GCM, for which QM is able to correct for important deficiencies in the intervariable dependencies. An example for Madrid (Fig.S3) shows that all RCMs perform equally well after QM.

[Fig.5]

**Figure 5: Two-dimensional Kernel density plots for the grid box over Warsaw. Blue histograms (and X-axis) refer to dew point temperature and red histograms (and Y-axis) refer to maximum temperature. The isolines for the observed WBGTp95 and WBGTp99 are also shown as the thick, dashed and solid black lines, respectively. Shadings represent the 2-D density distribution for the observations (first row), GCM (second row) and RCMs at EUR-11 (third to fifth rows). Very similar results are found for**
**EUR-44 (not shown). Contour lines represent the observed probabilities, which are overlaid the models probabilities for the sake of comparison. The numbers represent the Perkins skill score of distributional similarity (the closer to 1 the better).**

An overall conclusion about better performance is not evident since results depend on each grid box and GCM-RCM combination, and might be affected by compensations of biases in the individual variables. A summary for the evaluation of
the intervariable relationships across Europe is presented through the Perkins score (Fig.6). Lower scores are apparent in the raw GCM and RCM data, especially in areas with complex orography and south-eastern Europe. The two BC methods are able to improve the representation of the intervariable relationships in the whole continent, just by centering the distributions. High Perkins scores are found especially along the Atlantic coast. QM improves on ISIMIP in large areas, although low scores are found in Scandinavia (0.7-0.8) for the RCMs. The best results are obtained for GCM-QM, with large
scores also in northern Europe.

[Fig.6]

**Figure 6: Spatial distribution of Perkins skill scores calculated from the distribution of summer values of daily maximum temperature and daily mean dew point temperature, for the GCM (a-c) and the RCMs-EUR11 (d-l), for the raw (first column),**
**ISIMIP-corrected (second column) and QM-corrected (third column).**

### 3.3 Future changes of heat stress

For all the models considered (GCM, RCMs and BC methods) summer mean WBGT and WBGTp99 are projected to increase by the end of the 21st century under the RCP8.5 (Fig.7 and S4). For a given RCP, the major source of uncertainty in
the magnitude of this change comes from the choice of GCM or RCM, with a systematically lower change signal in the



RCMs. The differences in the climate change signal between the GCM and the RCMs may range between 0.5-1°C, depending on the RCM and RCP, for the European averaged values. It is related to the reduced summer warming in many EURO-CORDEX RCMs with respect to their driving GCMs that was noted already in previous works, which pointed to the different circulation patterns and surface energy fluxes and feedback mechanisms as possible causes for this (Keuler, et al.,

2016; Sørland, et al., 2018). The raw GCM projects changes in summer mean WBGT above 4.5°C over most parts of the continent with the highest values in the Alpine area (more than 6°C), whereas RCMs project increases between 3-5°C in most of the continent (Fig.7; shown are RCMs at 0.11° but similar results are found for 0.44°). The Alps and north of Scandinavia stand out with larger positive signals. By construction, ISIMIP approximately preserves the climate change signals of the input variables (Hempel, et al., 2013), whereas QM can potentially modify them (e.g. Gobiet, et al., 2015;

Ivanov, et al., 2018; see also Sect. 2.4). Our results show that little changes become apparent for the mean WBGT after ISIMIP (up to half a degree). The effect of the QM on the WBGT signal is especially noticeable for the case of the GCM, for which QM reduces the signal by up to 1.5°C and brings it closer to the RCM counterpart. The large, positive signal over the Alpine area is retained and stands out (although with smaller magnitude) for the RCMs-QM. Similar conclusions can be drawn for the change of WBGTp99, with a slightly more patchy spatial pattern (Fig.S4).

[Fig.7]

**Figure 7: Spatial distribution of changes in summer mean WBGT under RCP8.5 for the GCM (a-c) and RCMs-011 (d-l), for the raw (first column), ISIMIP-corrected (second column) and QM-corrected (third column). The climate change signals are calculated for the period 2070-2099 with respect to 1981-2010.**

The main conclusions hold qualitatively for the other RCPs, with quite consistent signals among RCMs and BC methods (Fig.8, upper panel). Differences between the GCM and RCMs projected signals are also evident for the input variables (Fig.8, middle and lower panels). These differences increase with the RCP and are larger for maximum temperature than for dew point temperature. Whereas the RCMs tend to lower the signal of the GCM for maximum temperature, they increase the

signal for dew point temperature. That is explained by the opposite behaviour of temperature and relative humidity, and the fact that models showing hotter temperatures tend to simulate lower relative humidity (Fischer & Knutti, 2013). In general, QM tends to expand the range of the raw RCM climate change signals and slightly lowers the median for the two CCLM simulations.

[Fig.8]

**Figure 8: Climate change signals for summer mean WBGT (upper panel), daily maximum temperature (Tx, central panel) and daily mean dew point temperature (Td, lower panel) for the period 2070-2099 with respect to 1981-2010 and RCPs 2.6, 4.5 and 8.5. Each box represents the changes across all grid boxes in Europe for the raw (blue), ISIMIP-corrected (orange) and QM-corrected (green) for the GCM and the RCMs (EUR-11 and EUR-44). Due to the different land-sea masks in the observations, GCM and**
**RCMs (EUR-44 and EUR-11), all boxplots consider the grid boxes common to all data sets.**





The modification of the climate change signal by BC is further analysed for the grid boxes over Warsaw (Fig.9a,b) and Madrid (Fig.9c,d), considering the change signals in the mean variables (left panels) and in the 99[th] percentile (right panel). In Warsaw, QM tends to reduce the signal of the GCM and RCMs towards lower maximum temperatures (Fig.9a). These changes in the signal are larger for the GCM than the RCMs. In this grid box, the effect of QM on mean dew point

temperature is negligible. As a consequence, the modification of the signal in mean WBGT is less than 0.5°C for the RCMs and 1°C for the GCM. Given that the projected change for mean WBGT is 3.5-5°C for the RCMs (~5.5°C for the GCM), the impact of the QM can amount to a maximum of 15% (18%) of the raw signal. Modifications in the climate change signal of the WBGTp99 by QM are smaller than for the mean, along with smaller changes in the signal of the 99th percentile of the input variables (Fig.9b; note that WBGTp99 is not necessarily linked to the 99th percentile of maximum temperature and

dew point temperature, but to some percentile in the upper tail of the distribution). In this example, however, it is evident that the preservation of trends by ISIMIP depends on the parameter under consideration, since e.g. the signal in the 99th percentile of dew point temperature for CCLM-044 is reduced by 1.3°C after the application of ISIMIP (see grey lines in Fig.9b). Again the modifications of the climate change signal are grid-box-specific, and negligible changes are found after QM and ISIMIP for the grid box closest to Madrid (Fig.9c,d).

[Fig.9]

**Figure 9: Scatter plots showing the effect of BC on the climate change signal for dew point temperature (X-axis), maximum temperature (Y-axis) and WBGT (coloured markers) for the grid box over Warsaw (a, b) and Madrid (c, d), RCP8.5 and the period 2070-2099 with respect to 1981-2010. The left panels show results for the change signal of the mean variables and the right**
**panels for the 99[th] percentiles. Each marker depicts results for a different data set (squares for the GCM, upwards triangles for the RCMs-044 and downwards triangles for RCMs-011). The black arrows point from the value in the raw data (thicker markers) to the change in the QM-corrected data, whereas the grey arrows point from the raw to the ISIMIP-corrected data (only discernible for the change signal of the 99[th] percentiles).**

## 4 Summary and Discussion

In the present work we compared global and regional climate model data at different spatial resolutions and bias-corrected by two bias correction methods (namely the ISIMIP method and empirical quantile mapping, QM) in order to assess the added value of 1) a more complex BC method and 2) bias-corrected RCM simulations versus bias-corrected GCM simulations, and 3) the role of downscaling and BC on the climate change signal of a multi-variate index. For this purpose we used GCM data from the CMIP5 HadGEM2-ES and the HadGEM-driven EURO-CORDEX simulations, at

approximately 12 km and 50 km horizontal resolution, respectively. The study was performed for the case of heat stress in Europe, considering as heat stress index the wet bulb globe temperature (WBGT) in shaded conditions. It depends on air temperature and dew point temperature, which were separately corrected prior to the index calculation. The performance of the models and methods in such a multi-variate framework was analysed. The results were examined considering present climate simulations (reference period 1981-2010) and future climate projections (2070-2099).





Regarding the performance of the two bias correction methods, the evaluation results show that both methods are able to correct for biases in the multi-variate WBGT as represented by the GCM and RCMs, with smaller biases for ISIMIP or QM depending on the GCM-RCM model chain. ISIMIP mostly retains the distributional features of the raw data, whereas QM narrows the two original distributions producing some improvement of the joint probability distribution with respect to

ISIMIP. The added value of higher climate model resolution (from GCM to RCM and from EUR-44 to EUR-11) is not evident in the evaluation of the bias-corrected WBGT statistics, since the biases of both the GCM and RCMs become indistinguishable after bias correction. The joint probabilities are, however, better reproduced by the RCMs after the two bias corrections, especially due to a more accurate representation of these relationships in the raw data. For those cases (i.e. grid boxes) for which the raw models do not well represent the intervariable relationships (e.g. CCLM for the grid box closest to

Warsaw) some biases in the joint distribution may remain after bias correction. Large biases of the raw GCM in the intervariable dependencies might be related to biases in large-scale processes and feedbacks. Further research is needed to understand the causes for these biases, while the application of BC for those cases in a multi-variate context is then debatable (Piani, et al., 2010; Ehret, et al., 2012; Muerth, et al., 2013). Other methods and approaches (i.e. perfect prognosis approach, high resolution regional models) are viable alternatives to bias correction in those cases (Maraun, 2016; Maraun, et al.,

15 2017).

Regarding climate change projections of WBGT, the largest differences for a given RCP come from the use of GCM versus RCM data, with systematically lower signals for the RCMs. The GCM-RCM differences amount to 0.5-1°C for the European averaged signal and increase with the emission scenario, regardless of the bias correction method and RCM resolution. QM tends to reduce the signal in both GCM and RCMs, bringing the GCM-based and RCM-based results closer to each other.

Some modifications of the raw RCM signal are visible after QM (up to 20% of the raw signal), however the original signal of the GCM is qualitatively retained by the RCMs-QM, with larger increments in the Alpine ridge and north Scandinavia. Although the ISIMIP method is by construction a trend-preserving BC method, due to the non-linearities in the WBGT calculation some modifications of the signal in WBGT statistics may become apparent after the correction. The modifications of the climate change signals due to bias correction are generally smaller than the model uncertainty (spread

over the GCM and RCMs at two resolutions) by the end of the century. The magnitude of these changes should be also analysed in the context of natural variability (Räisänen, 2001), since the latter can mask or enhance long-term trends.

Summarizing, there is some added value of QM with respect to ISIMIP in the representation of the intervariable structures, whereas the present-climate evaluation shows limited added value of bias-corrected RCM versus bias-corrected GCM data. More distinct results between RCMs and GCM are obtained regarding climate projections, with systematically smaller

change signals in the RCMs. The bias-corrected data qualitatively retain the change signal of the raw counterparts, although QM tends to decrease the signal of the WBGT and the input variables.

Some limitations and points for discussion remain. The use of a single GCM (even downscaled with several RCMs and bias-corrected) for the production of climate projections does not sample the full uncertainty range and the use of large ensembles of simulations is strongly recommended. GCMs typically produce larger estimates for the change signal of temperature than





RCMs due to different circulation patterns and surface energy fluxes and feedback mechanisms (Keuler, et al., 2016; Sørland, et al., 2018). HadGEM in particular projects an increase of summer mean WBGT of 5°C (European-average, end of 21st century, RCP8.5) and HadGEM-driven RCM simulations of about 4-4.5°C. These results are at the upper limit of the uncertainty range when compared to a large ensemble of GCM-RCM simulations (Casanueva, et al., 2018b). Therefore
relying only on this GCM could lead to misleading conclusions when combined with other factors in impact assessments. This example highlights even further the need for ensembles of simulations.

The differences between the two RCM resolutions were negligible in our study, mainly due to the experimental design (both resolutions are remapped onto the 50x50km observational grid). The higher resolution RCMs could show some potential added value if the evaluation would be carried out at their original resolution. However, there is no pan-European, high-
resolution, observational grid neither for air temperature nor for dew point temperature (or relative humidity) to bias-correct and evaluate these simulations. While high-resolution grids for temperature are available at a national level, the lack of a gridded product for relative humidity remains a limitation. Furthermore, model evaluation can depend on the reference data set employed and observations play a fundamental role in bias correction, especially in QM for which the whole distribution is adjusted. Previous studies have shown that model uncertainty dominates over observational uncertainty for the case of
mean temperature (Kotlarski, et al., 2017), but dew point temperature (or relative humidity) has not been broadly investigated so far. In the present work, we do not account for observational uncertainty but acknowledge that the reliability and spatial representativeness of the reference data set might quantitatively modify the results. Future works including also a comparison of different observational data products might shed light on the robustness of the current results.

*Code availability*
The code used for ISIMIP is an open source, R package *climate4R* (Cofiño, et al., 2018; Iturbide, et al., 2019) available from a GitHub repository (https://github.com/SantanderMetGroup/downscaleR). The quantile mapping code is also an R package that can be obtained from the authors upon request. The heat stress index is calculated with the R package *HeatStress* (https://github.com/anacv/HeatStress). All the code to performed derived analyses, calculations and plots is also based on R
scripts, which are available upon request.

*Data availability*
The model simulations (EURO-CORDEX RCMs and HadGEM2-ES) used in this study are accessible via the Earth System Grid Federation (ESGF archive, https://esgf.llnl.gov). The ESGF architecture consists of a global system of distributed
nodes, which interoperate with other according to a peer-to-peer paradigm, i.e. each node can act as the provider or the consumer of services; they can join or leave the federation dynamically, without affecting the operations of the other nodes. The user needs to have an OpenID and can select different search criteria. To get the RCM data used in this work the selection is: Project= 'CORDEX'; Domain= 'EUR-11' and 'EUR-44'; Experiment= 'historical', 'rcp26', 'rcp45', 'rcp85'; Time Frequency= 'day'; Variable= 'tas', 'tasmax', 'hurs' (or 'huss' and 'ps'). The HadGEM2-ES data are also available





through the ESGF System, selecting the search criteria: Project= 'CMIP5'; Institute= 'MOHC'; Model= 'HadGEM2-ES'; Experiment= 'historical', 'rcp26', 'rcp45', 'rcp85'; Time Frequency= 'day'; Ensemble= 'r1i1p1'; Variable= 'tas', 'tasmax', 'huss', 'psl'.

*Author contributions*

AC, SK and TK conceived the study. SH implemented the ISIMIP method in R and SK re-edited the QM code used in this work. AC performed the analyses and SK and SH gave support regarding the bias correction methods and model data. AMF and CS provided ideas for new analyses and illustrations. AC wrote the first draft of the manuscript and all authors reviewed the text and contributed to the final version.

*Competing interests*

The authors declare that they have no conflict of interest.

*Acknowledgements*

The authors are grateful to the World Climate Research Programme's Working Group on Regional Climate, and the Working Group on Coupled Modelling, former coordinating body of CORDEX and responsible panel for CMIP5. We also thank the climate modelling groups for producing and making available their model output. We also acknowledge the Earth System Grid Federation infrastructure an international effort led by the U.S. Department of Energy's Program for Climate Model Diagnosis and Intercomparison, the European Network for Earth System Modelling and other partners in the Global
Organisation for Earth System Science Portals (GO-ESSP). We also thank Curdin Spirig (ETH Zurich) for preprocessing the simulation data and Dr. Jan Rajczak (ETH Zurich) for providing an earlier version of the quantile mapping code. We also thank our HEAT-SHIELD colleagues (https://www.heat-shield.eu/) and the CH2018 group (www.ch2018.ch) for technical support and scientific discussions. Financial support for this work is provided by the HEAT-SHIELD Project (HORIZON 2020, research and innovation programme under the grant agreement 668786). The authors wish to thank the Swiss National
Supercomputing Centre (CSCS) for providing the technical infrastructure.

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





**List of tables**

| RCM | EUR-11 | | | EUR-44 | | | Reference |
|---|---|---|---|---|---|---|---|
| | **RCP2.6** | **RCP4.5** | **RCP8.5** | **RCP2.6** | **RCP4.5** | **RCP8.5** | |
| RACMO22E | X | X | X | X | X | X | Meijgaard, et al., 2008 |
| RCA4 | | X | X | X | X | X | Samuelsson, et al., 2011 |
| CCLM4-8-17 | | X | X | | | X | Rockel, et al., 2008 |
| Total # per RCP | 1 | 3 | 3 | 2 | 2 | 3 | |

**Table 1: EURO-CORDEX RCMs driven by HadGEM, for the two spatial resolutions (EUR-11 for approximately 12km spatial resolution and EUR-44 for approximately 50km resolution) and three RCPs (RCP2.6, RCP4.5 and RCP8.5).**





**List of figures**

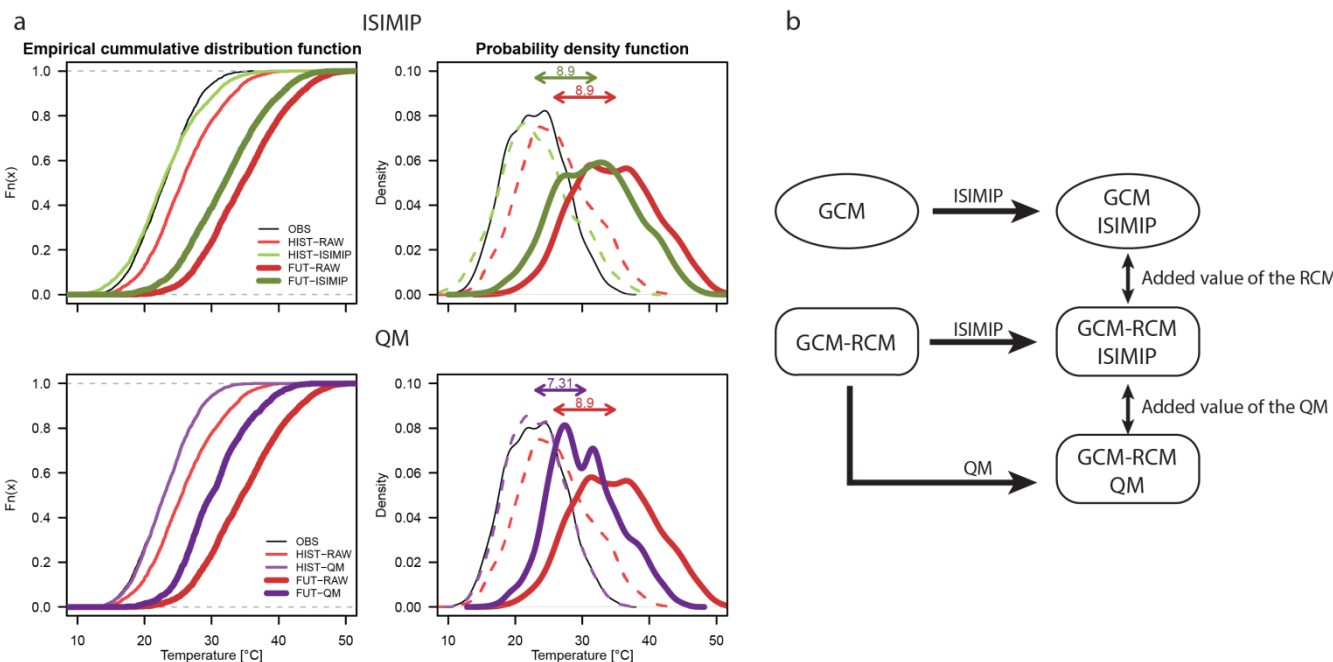

10 **Figure 1: (a) Example of the effect of the two bias correction methods on the empirical cumulative distribution function (left) and the probability density function (right). Observations are depicted in black and historical (HIST) and future (FUT) model simulations in light and dark colours, respectively. Raw data are depicted in red, ISIMIP-corrected data in green (upper panel) and QM-corrected data in purple (lower panel). The magnitude of the mean change signal is shown with the arrows. This example corresponds to daily maximum temperature as represented by HadGEM for an exemplary grid box (HIST: 1981-2010, FUT:**
15 **2070-2099 for RCP8.5). (b) Conceptual scheme of the present study.**



**Figure 2: Biases of mean (first row), 95th percentile (second row) and 99th percentile (third row) of summer WBGT for the GCM and RCMs at EUR-44 and EUR-11. Biases are calculated as model minus observations. Each box represent the biases across all grid boxes for the raw (blue), ISIMIP-corrected (orange) and QM-corrected (green). Due to the different land-sea masks in the observations, GCM and RCMs (EUR-44 and EUR-11), all boxplots consider the grid boxes common to all data sets.**

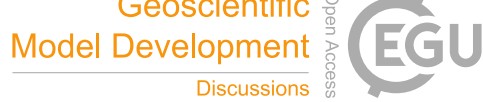



**Figure 3: Spatial distribution of the observed 99th percentile of summer WBGT (WBGTp99, panel a) and model biases for the GCM (b-d) and RCMs-EUR11 (e-m), for the raw (first column), ISIMIP-corrected (second column) and QM-corrected (third column). The grid boxes over Warsaw and Madrid are marked as reference for subsequent analyses.**

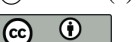



**Figure 4: Intervariable relationship for the observations and (raw and bias corrected) model data, for RACMO-011 (a-c) and**
10 **CCLM-011 (d-f) for the grid box over Warsaw. Each scatter plot represents pairs of values of daily dew point temperature (X-axis) and maximum temperature (Y-axis) which produce summer WBGT values above WBGTp95. The three coloured markers correspond to WBGT values for the observations (circles in all panels), raw RCMs (squares in a,d), RCM-ISIMIP (downward triangles in b,e) and RCM-QM (upward triangles in c,f). Isolines also represent WBGT values and the thicker line depicts the observed WBGTp99.**



**Figure 5: Two-dimensional Kernel density plots for the grid box over Warsaw. Blue histograms (and X-axis) refer to dew point temperature and red histograms (and Y-axis) refer to maximum temperature. The isolines for the observed WBGTp95 and**



**WBGTp99 are also shown as the thick, dashed and solid black lines, respectively. Shadings represent the 2-D density distribution for the observations (first row), GCM (second row) and RCMs at EUR-11 (third to fifth rows). Very similar results are found for EUR-44 (not shown). Contour lines represent the observed probabilities, which are overlaid the models probabilities for the sake of comparison. The numbers represent the Perkins skill score of distributional similarity (the closer to 1 the better).**





**Figure 6: Spatial distribution of Perkins skill scores calculated from the distribution of summer values of daily maximum temperature and daily mean dew point temperature, for the GCM (a-c) and the RCMs-EUR11 (d-l), for the raw (first column), ISIMIP-corrected (second column) and QM-corrected (third column).**



**Figure 7: Spatial distribution of changes in summer mean WBGT under RCP8.5 for the GCM (a-c) and RCMs-011 (d-l), for the raw (first column), ISIMIP-corrected (second column) and QM-corrected (third column). The climate change signals are calculated for the period 2070-2099 with respect to 1981-2010.**



5 **Figure 8: Climate change signals for summer mean WBGT (upper panel), daily maximum temperature (Tx, central panel) and daily mean dew point temperature (Td, lower panel) for the period 2070-2099 with respect to 1981-2010 and RCPs 2.6, 4.5 and 8.5. Each box represents the changes across all grid boxes in Europe for the raw (blue), ISIMIP-corrected (orange) and QM-corrected (green) for the GCM and the RCMs (EUR-11 and EUR-44). Due to the different land-sea masks in the observations, GCM and RCMs (EUR-44 and EUR-11), all boxplots consider the grid boxes common to all data sets.**



**Figure 9: Scatter plots showing the effect of BC on the climate change signal for dew point temperature (X-axis), maximum temperature (Y-axis) and WBGT (coloured markers) for the grid box over Warsaw (a, b) and Madrid (c, d), RCP8.5 and the period 2070-2099 with respect to 1981-2010. The left panels show results for the change signal of the mean variables and the right panels for the 99[th] percentiles. Each marker depicts results for a different data set (squares for the GCM, upwards triangles for the RCMs-044 and downwards triangles for RCMs-011). The black arrows point from the value in the raw data (thicker markers) to the change in the QM-corrected data, whereas the grey arrows point from the raw to the ISIMIP-corrected data (only discernible for the change signal of the 99[th] percentiles).**