# Peer review of "Climate projections of a multi-variate heat stress index: the role of downscaling and bias correction"

_Geoscientific Model Development, 2018_

## Referee Comment (RC1) · Anonymous Referee #1 · 22 Mar 2019

The manuscript investigates the role of downscaling and bias correction to capture the climate change signal of multi-variate heat stress index, by comparing GCM and RCM simulations at different spatial resolutions. The corrected heat stress index (WBGT in the shade conditions) is calculated from air temperature and dew point temperature, which were separately corrected using two BC methods; a) ISIMIP (parametric quantile mapping) and b) empirical quantile mapping. The bias-correction methods applied in the manuscript are not newly developed techniques. However, the application on a multi-variate index and the evaluation of the corrected index are a needed task in the topic of bias-correction on climate model simulations. The overall manuscript is well written, and most of the figures included are clearly stated.

[Figure]

Specific comments:

- Page 3, line 19: More explanation on "intensity-dependent biases" would help of the quantile mapping. Can you provide a reference for the term?

- Page 5, line 3: I am curious about the reasoning of using daily 'mean' dew point temperature, instead of using daily maximum dew point temperature, to calculate the daily maximum WBGT.

- Page 9, line 30: I like joint distributions of two input variables in Fig 5 to understand the characteristics of joint dependency for climate simulations better. However, it would be good to see some statistics like the correlation to show dependence between two input variables, maximum temperature and dew point temperature. In Fig 4d, it seems there exists a stronger negative correlation between two variables in the raw CCLM, compared to the correlation in Obs. If the negative relationship is stronger on extremes (e.g., above 95th percentile) of two variables, that might bring inaccurate bias adjustment in QM, leading to the underestimated negative biases?

- Page 11, line 24-25: I don't know how the conclusion is drawn. By comparing average Perkins scores?

- Page 15, line 6: If I understand correctly, you used a single ensemble (r1i1p1) of HadGEM2-ES. Do the biases relate to the biases across ensemble runs? If we use more ensemble members of the HadGEM2 simulation, do we expect the smaller biases?

- Fig 1a: I am a bit confused. Are the CDFs of the (historical and future) RAW from RCM? Or GCM?

---

## Referee Comment (RC2) · Anonymous Referee #2 · 6 Apr 2019

This study examines simulations of a climate indicator over Europe with implications for human health (heat stress index, Wet Bulb Globe Temperature (WGBT)). Bias corrected simulations from both Global and Regional Climate Models (GCMs and RCMs) are compared with the goal of determining the added value provided by the RCM in this scenario as well as more complex BC methods (QM vs ISIMIP). One novel aspect of this study in particular is the fact that the WBGT is multi-variate as it is based on both temperature and dew point temperature, which adds considerable complexity in the context of assessing the value of bias correction methods due to intervariable relationships. Overall, the manuscript is very clear, concise, and provides some evidence to support its conclusions, in particular that the chosen RCMs added little value with

respect to the GCM after bias correction. The authors have properly acknowledged some major caveats to this conclusion, including (1) Only 1 GCM was used in the comparison between RCMs and (2) Regridding the high-resolution RCM simulations to the much coarser reference dataset may reduce any potential added value they would have otherwise provided. These open up several avenues for future work.

Specific Comments:

- Page 5, Line 31: Given the issues you had to account for due to the 360-day calendar in HadGEM-ES, why did you select it for this study over other CMIP5 GCMs which have more standard calendars?

- Page 6, Lines 4-6: Could you also be more specific in regards to what beneficial features aren't smoothed out from the high-resolution simulations after regridding?

- Page 11, Lines 20-25: Some interpretations which explain these results would be nice to have here, in particular to explain the lower skill in Scandinavia for the RCMs. It might be helpful to see some additional maps showing the standard deviations of daily max temperature and daily mean dewpoint temperature.

- Page 14, Lines 27-28: This would be a bit beyond the scope of this paper, but given that the RCMs chosen in this study are still coarse enough to require many parameterizations, I would be interested in seeing future work examine the robustness of this conclusion for convection permitting models.

―――――――――――――――――

---

## Author Response (AR1)

*Dear Ana Casanueva,*

*We are pleased to inform you that the open discussion of your following manuscript has been closed:*
*Journal: GMD*
*Title: Climate projections of a multi-variate heat stress index: the role of downscaling and bias correction*
*Author(s): Ana Casanueva et al.*
*MS No.: gmd-2018-294*
*MS Type: Model evaluation paper*

*No more referee comments and short comments will be accepted. Now the public discussion shall be completed as follows:*
*You - as the contact author - are requested to respond to all referee comments (RCs) by posting final author comments on behalf of all co-authors no later than 09 May 2019 (final response phase) at: https://editor.copernicus.org/gmd-2018-294/final-response*
*After your posts, you have to explicitly finalize the final-response form before you are asked in a separate email to prepare and submit your revised manuscript for peer-review completion and potential final publication in GMD.*
*When replying to the referee comments (RCs) it is sufficient to post one author comment (AC) by starting a new discussion thread. Please also consider replying to short comments (SCs) from the scientific community. The response to the Referees shall be structured in a clear and easy-to-follow sequence: (1) comments from Referees, (2) author's response, (3) author's changes in manuscript.*
*Preparation and submission of a revised manuscript is encouraged only if you can satisfactorily address all comments and if the revised manuscript meets the high quality standards of GMD (https://www.geoscientific-model-development.net/peer_review/review_criteria.html). In case of doubt, please ask the handling Topical Editor directly whether they would encourage submission of a revised manuscript or not.*
*Please note also that the submission of a revised manuscript does not ensure publication in GMD. The Topical Editor will carefully assess your revised manuscript in view of the interactive public discussion and may forward it to the original or new Referees for further commenting.*
*To log in, please use your Copernicus Office user ID 160310.*
*You are invited to monitor the processing of your manuscript via your MS Overview: https://editor.copernicus.org/GMD/my_manuscript_overview*
*Thank you very much in advance for your cooperation. In case any questions arise, please do not hesitate to contact me.*

*Kind regards,*
*Natascha Töpfer*
*Copernicus Publications*
*Editorial Support*
*editorial@copernicus.org*
*on behalf of the GMD Editorial Board*

Dear Editor,

Many thanks for the guidelines and constructive comments to our manuscript. We now present our revised manuscript and our replies to the reviewers' comments and suggestions. We appreciate the work of the editor and the referees in helping us to improve the manuscript. Following one of the reviewers' recommendation, we obtained the correlation between air temperature and dew point temperature (all series and for pairs of values producing extreme heat stress) and those results have been discussed and included in Figs.4,5. Also maps of the standard deviation of the input variables have been included in the supplementary material (Figs.S4, S5), in order to discuss this aspect together with the Perkins scores, as the second reviewer suggested. A short section about evaluation metrics have been included in the methodology, for the sake of clarity.

Please find below the point-by-point responses to the reviewers' comments and the new manuscript with (and without) highlighted changes. We hope the revised manuscript is now acceptable for publication in *Geoscientific Model Development*. All authors agree on the current form of the manuscript.

Dr. Ana Casanueva, on behalf of the authors.

**Reviewer #1:**

*The manuscript investigates the role of downscaling and bias correction to capture the climate change signal of multi-variate heat stress index, by comparing GCM and RCM simulations at different spatial resolutions. The corrected heat stress index (WBGT in the shade conditions) is calculated from air temperature and dew point temperature, which were separately corrected using two BC methods; a) ISIMIP (parametric quantile mapping) and b) empirical quantile mapping. The bias-correction methods applied in the manuscript are not newly developed techniques. However, the application on a multi-variate index and the evaluation of the corrected index are a needed task in the topic of bias-correction on climate model simulations. The overall manuscript is well written, and most of the figures included are clearly stated.*

**Response:** We thank the reviewer for the comments and the time devoted to our paper. Please, see below our point-by-point responses and the changes highlighted in the new version of the manuscript.

*Specific comments:*
*- Page 3, line 19: More explanation on "intensity-dependent biases" would help of the quantile mapping. Can you provide a reference for the term?*

Thanks for the comment. A brief explanation and a reference have been included in the revised manuscript.

"Quantile mapping is, by construction, able to correct for intensity-dependent biases (i.e. biases that change throughout the distribution, Gobiet et al. 2015)"

*- Page 5, line 3: I am curious about the reasoning of using daily 'mean' dew point temperature, instead of using daily maximum dew point temperature, to calculate the daily maximum WBGT.*

Thanks for raising this point. The reason for approximating daily maximum heat stress using daily mean dew point temperature is the non-availability of data at hourly resolution in most observation-based and simulated datasets used in the current work (ideally, data at hourly or even higher temporal resolution should be used). Unlike relative humidity, which is anticorrelated with air temperature and changes strongly along the day, dew point temperature only slightly varies during the day. The following figure shows the diurnal cycle of air temperature (Ta), dew point temperature (Td) and WBGT for a typical summer day in Lugano (Switzerland), in the period 1981-2010. A similar result was found for other Swiss locations, where hourly data were available. In those cases dew point temperature shows a diurnal range of approximately 1-1.5°C.

[Figure]

Given the small diurnal cycle, daily mean values of dew point temperature in combination with daily maximum air temperature were used in the present work to approximate daily maximum heat stress. Also note that daily mean dew point temperature was obtained from

daily mean temperature and relative humidity in models and observations (neither hourly nor minimum relative humidity data were available from either models or observations).

*- Page 9, line 30: I like joint distributions of two input variables in Fig 5 to understand the characteristics of joint dependency for climate simulations better. However, it would be good to see some statistics like the correlation to show dependence between two input variables, maximum temperature and dew point temperature. In Fig 4d, it seems there exists a stronger negative correlation between two variables in the raw CCLM, compared to the correlation in Obs. If the negative relationship is stronger on extremes (e.g., above 95th percentile) of two variables, that might bring inaccurate bias adjustment in QM, leading to the underestimated negative biases?*

Thanks for the comment. The following table summarizes the (Pearson) correlation coefficients between daily maximum temperature and daily mean dew point temperature, considering the full series (left) and the pairs of values that produce WBGT above the 95th percentile:

| | Full series (JJA) | | | Pairs producing WBGT>WBGTp95 | | |
|---|---|---|---|---|---|---|
| **OBS.** | **0.54** | | | **-0.55** | | |
| | RAW | ISIMIP | EQM | RAW | ISIMIP | EQM |
| GCM | 0.16 | 0.2 | 0.16 | -0.71 | -0.61 | -0.71 |
| RACMO-044 | 0.49 | 0.52 | 0.49 | -0.72 | -0.65 | -0.71 |
| **RACMO-011** | **0.42** | **0.46** | **0.42** | **-0.43** | **-0.53** | **-0.45** |
| RCA-044 | 0.23 | 0.31 | 0.3 | -0.74 | -0.71 | -0.8 |
| RCA-011 | 0.27 | 0.35 | 0.34 | -0.67 | -0.62 | -0.7 |
| CCLM-044 | 0.08 | 0.16 | 0.14 | -0.85 | -0.81 | -0.88 |
| **CCLM-011** | **0.02** | **0.09** | **0.06** | **-0.82** | **-0.83** | **-0.88** |

In general air and dew point temperatures present a positive, linear relation (r=0.54). However, extreme values of WBGT are produced (at this specific grid box close to Warsaw) under high values of air temperature and low values of dew point temperature, or vice versa (r=-0.55). RACMO stands out as the best of the three RCMs representing the intervariable relationships, for the full series (see also Fig.5, third row) and the highest WBGT (see also Fig.4a-c). However, for the GCM and CCLM the two full series do not correlate linearly (r is approximately 0) and they are too strongly anticorrelated for the extreme WBGT (see also Fig.5 last row and Fig.4d-f for CCLM).

The two bias correction methods do not tackle the temporal correlation and maintain the temporal structure of the raw data and the temporal correlation between air and dew point temperatures remain similar to the raw counterpart. A slightly stronger negative correlation for the pairs producing extreme WBGT is obtained for the CCLM simulations after QM. That means that high values of dew point temperature would then be linked to rather low air temperatures (stronger negative correlation than for the observations), which may imply lower values for extreme WBGT. This together with an overcorrection of the positive bias in extreme air temperatures (Fig.4d,f) might favour negative biases WBGTp99.

Some further explanations and the correlation coefficients from the table above have been included in Figs.3 (for pairs of variables producing WBGT>WBGTp95) and 4 (all pairs) in the revised manuscript.

*- Page 11, line 24-25: I don't know how the conclusion is drawn. By comparing average Perkins scores?*

This conclusion is drawn from Fig.6, where the spatial distribution of Perkins scores is depicted. In particular, the best results for GCM-QM are found in Fig.6c, with values close to 1 for all Europe. It is explicitly mentioned in the revised manuscript that this conclusion is shown in the mentioned plot.

*- Page 15, line 6: If I understand correctly, you used a single ensemble (r1i1p1) of HadGEM2-ES. Do the biases relate to the biases across ensemble runs? If we use more ensemble members of the HadGEM2 simulation, do we expect the smaller biases?*

All results are based on a single ensemble r1i1p1 of HadGEM-ES. When mentioning the need of large ensembles of simulations, we refer to ensembles built on different GCMs. A larger ensemble of GCM-RCMs (as in Casanueva et al. 2018 for climate projections of heat stress), can ease the quantification of the robustness and uncertainties in the projections. Using large ensembles, considering also other HadGEM2 runs, does not necessarily mean a reduction of model biases, but their own biases.

*- Fig 1a: I am a bit confused. Are the CDFs of the (historical and future) RAW from RCM? Or GCM?*

Thanks for the comment. This figure attempts to illustrate the generic bias correction procedure for any (regional or global) model. The numbers correspond to HadGEM-ES (i.e. GCM). The caption has been changed for the sake of clarity.

**Reviewer #2:**

*This study examines simulations of a climate indicator over Europe with implications for human health (heat stress index, Wet Bulb Globe Temperature (WGBT)). Bias corrected simulations from both Global and Regional Climate Models (GCMs and RCMs) are compared with the goal of determining the added value provided by the RCM in this scenario as well as more complex BC methods (QM vs ISIMIP). One novel aspect of this study in particular is the fact that the WBGT is multi-variate as it is based on both temperature and dew point temperature, which adds considerable complexity in the context of assessing the value of bias correction methods due to intervariable relationships. Overall, the manuscript is very clear, concise, and provides some evidence to support its conclusions, in particular that the chosen RCMs added little value with respect to the GCM after bias correction. The authors have properly acknowledged some major caveats to this conclusion, including (1) Only 1 GCM was used in the comparison between RCMs and (2) Regridding the high-resolution RCM simulations to the much coarser reference dataset may reduce any potential added value they would have otherwise provided. These open up several avenues for future work.*

**Response:** We thank the reviewer for the comments and the time devoted to our paper. Please, see below our point-by-point responses and the changes highlighted in the new version of the manuscript.

*Specific Comments:*
*- Page 5, Line 31: Given the issues you had to account for due to the 360-day calendar in HadGEM-ES, why did you select it for this study over other CMIP5 GCMs which have more standard calendars?*

The motivation of the present work started after Kjellstrom et al. 2017, who estimated population heat exposure and impacts on working people at a global scale with GCM data from the ISIMIP project. Only two out of the 4-5 ISIMIP-corrected GCMs were used in the cited work, as representative of the range of different models used by IPCC for global temperature change (GTC) estimates; HadGEM2 producing GTC results close to the upper limit of models and GFDL producing results close to the lower limit. Thus our aim was to assess the robustness of those results based on GCM data and the ISIMIP correction. Unfortunately, GFDL was only dynamically downscaled through the RCA4 (SMHI) regional climate model within EURO-CORDEX, whereas HadGEM2 provided the boundary and initial conditions for three RCMs (at two spatial resolutions and for three scenarios, see Table 1), therefore only the latter was considered. In general, however, we believe that the filling in of a few missing values in order to account for the full Gregorian calendar does not distort the results obtained.

Kjellstrom, T., Freyberg, C., Lemke, B., Otto, M. and Briggs, D.: Estimating population heat exposure and impacts on working people in conjunction with climate change. International Journal of Biometeorology, 01 3, 62, pp. 291-306, 2018.

*- Page 6, Lines 4-6: Could you also be more specific in regards to what beneficial features aren't smoothed out from the high-resolution simulations after regridding?*

Thanks for the comment. That part of the sentence refers to the aspects mentioned before. We mean that the added value of the high resolution on certain processes might still be evident after regridding/smoothing. The sentence has been rewritten to:

"As a consequence, there will be aspects of the added value of the high-resolution EUR-11 experiments (related to better-resolved, fine-scale processes; Prein, et al., 2015) that can be smoothed out, but **they** may still be present after remapping them onto a coarse resolution (Casanueva et al. 2016)."

As an example, the following figure shows daily mean precipitation (period 1989-2008) for the Alpine region as represented by a single RCM (CCLM) at the 12km original resolution, regridded onto the 50km (12kmAGG) and the original simulation at 50km. Compared to the latter, the aggregated 12km version shows more details that are also present in the full 12km version. See more details of this added value analysis in Casanueva et al. 2016.

[Figure]

Casanueva, A., Kotlarski, S., Herrera, S., Fernández, J., Gutiérrez, J. M., Boberg, F., Colette, A., Christensen, O. B., Goergen, K., Jacob, D., Keuler, K., Nikulin, G., Teichmann, C. and Vautard, R.: Daily precipitation statistics in a EURO-CORDEX RCM ensemble: added value of raw and bias-corrected high-resolution simulations. Climate Dynamics, 47, pp. 719-737, 2016.

*- Page 11, Lines 20-25: Some interpretations which explain these results would be nice to have here, in particular to explain the lower skill in Scandinavia for the RCMs. It might be helpful to see some additional maps showing the standard deviations of daily max temperature and daily mean dewpoint temperature.*

Thanks for the comment. The two figures below show the standard deviation of the two variables. The areas with larger biases in the standard deviation agree with those with smaller Perkins scores in Fig.6, pointing out that deficiencies in the temporal variability of the individual variables might be responsible for some of the deficiencies in the intervariable relationships. It is also noticeable that the lower skill in Scandinavia after QM corresponds to lower variability (standard deviation) than observed in the two variables.

These two plots have been added to the Supplementary Material and the text has been completed as follows:

"High Perkins scores are found especially along the Atlantic coast. QM improves on ISIMIP in large areas, although low scores are found in Scandinavia (0.7-0.8) for the RCMs. The spatial distribution of the scores agrees qualitatively with biases in the temporal variability of maximum and dew point temperatures (Figs.S4-S5). This is a first order indication that the misrepresentation of the temporal variability of the individual variables might be responsible for most of the deficiencies in the intervariable relationships. Raw model data overestimate the temporal variability especially in Eastern Europe, leading to Perkins scores lower than 0.6. In other areas, such as Scandinavia, the models underestimate the temporal variability of the two input variables, and thus present the lowest scores even after QM."

[Figure]

Standard deviation of the distribution of daily maximum temperatures (1981-2010, JJA). Results for the observations (a), GCM (b-d) and three RCMs-EUR11 (e-m). Raw and bias-corrected data are depicted in columns.

[Figure]

As the previous figure, but for daily mean dew point temperature.

*- Page 14, Lines 27-28: This would be a bit beyond the scope of this paper, but given that the RCMs chosen in this study are still coarse enough to require many parameterizations, I would be interested in seeing future work examine the robustness of this conclusion for convection permitting models.*

Thanks for the comment. That is certainly an interesting point for future work. A sentence on this has been included in the discussion:
"Future works including convection-permitting simulations could help to assess the robustness of these results".

---

## Author Response (AR2)

Dear Ana Casanueva,

Many thanks for submitting a revised version of your manuscript addressing the open discussion comments. Based on the referee report I have received, I am satisfied that all the scientific points raised in the review have been adequately addressed.

However, before I can accept this manuscript for publication, I must request that you fulfill our Code Availability policy. In particular, all the code associated with the manuscript should be made available form an archive with a DOI, for example from the ZENDO (or equivalent) data repository. Please see of Code and data policy for full details. I seem to recall that you were arranging this for the Quantile Mapping software. A DOI is important to ensure that the link is permanent (indeed, code repositories can be deleted) and links to the version of the software described/used in the paper. I would also ask that you write the version of the R libraries used in your "Code availability section". Please let me know if you require further guidance on this.

Best regards,

Lauren Gregoire

Dear Editor,

Many thanks for the comments to our manuscript. We appreciate the work of the editor and the referees in helping us to improve the manuscript. We arranged the DOIs for all the code used in the paper and they were included in the "Code availability" section. Software and packages' versions have been also mentioned.

Please find attached the new manuscript with (and without) highlighted changes. We hope the revised manuscript is now acceptable for publication in *Geoscientific Model Development*. All authors agree on the current form of the manuscript.

Dr. Ana Casanueva, on behalf of the authors.

[revised manuscript text omitted]